# Is the Superparamagnetic Approach Equal to Radioisotopes in Sentinel Lymph Node Biopsy? The Over-Collecting Node Issue in Breast Cancer Patients

**DOI:** 10.3390/jcm14093148

**Published:** 2025-05-01

**Authors:** Karolina Zalewska, Maria Skonieczna, Dariusz Nejc, Piotr Pluta

**Affiliations:** 1Department of Surgical Oncology, Medical University of Lodz, 90-419 Lodz, Poland; klinikacho@csk.umed.pl (K.Z.); dareknejc@op.pl (D.N.); 2Department of Surgical Oncology and Breast Diseases, Polish Mother’s Memorial Hospital—Research Institute in Lodz, 93-338 Lodz, Poland; sek21@iczmp.edu.pl

**Keywords:** breast cancer, sentinel lymph node biopsy, SLNB, superparamagnetic iron oxide, radioisotope

## Abstract

**Background:** Sentinel lymph node biopsy (SLNB) is the primary procedure for nodal assessment in breast cancer patients. Radioisotopes (RIs) are considered the gold-standard tool. The ferromagnetic technique (superparamagnetic iron oxide—SPIO) is a non-isotope alternative SLNB method. This study compares the efficacy of SPIO and RI SLNB across two independent breast cancer centres. **Methods:** A total of 406 breast cancer patients, who underwent SLNB between January 2021 and December 2022, were analysed (SPIO—223 patients, RI—183 patients). Statistical tests, including Mann–Whitney U and chi-squared analyses, compared the SLN identification rates, the number of SLNs retrieved, and the positive node detection rates. **Results:** The identification rates were similar for SPIO and RI (two-tailed Fisher’s exact test, *p* = 1.0). The SPIO method retrieved significantly more SLNs than RI (3.26 vs. 2.15; *p* < 0.001). A larger proportion of patients in the SPIO group had ≥ 5 SLNs removed (20.2% vs. 8.7%; *p* = 0.001). A statistically significant difference was observed in the proportion of metastatic SLNs to harvested SLNs between the techniques, with a larger proportion detected in the RI group (1/7.88) compared to the SPIO group (1/14.81) (chi-squared test, *p* < 0.03). **Conclusions:** In our study, the SPIO and RI methods effectively collected SLNs. The gold-standard RI method offers distinct advantages, including its precise and consistent dosing unaffected by patient-specific factors and a highly targeted approach to node identification. These features minimise the risk of over-dissection and ensure that only the most clinically relevant nodes are removed. We note that the SPIO technique in SLNB in breast cancer patients requires further standardisation.

## 1. Introduction

Breast cancer is the most frequently reported malignant neoplasm in women worldwide. This trend is also observed in Poland [1]. Sentinel lymph node biopsy (SLNB) is an essential part of surgical management in patients with breast cancer. The sentinel lymph node (SLN) is one of the first nodes receiving lymphatic drainage from the primary tumour. The observation of the metastases spreading through the lymphatic route resulted in the development of SLNB. This procedure establishes the severity of the disease and determines further treatment options. It has replaced axillary lymph node dissection (ALND), being equivalent in axillary staging, with comparable results in terms of disease-free survival (DFS), overall survival (OS), and recurrence rates (RRs). At the same time, it is associated with lower postoperative morbidity [2].

SLNB is indicated for patients with a cN0 status. This means that the procedure is reserved for cases where the physical examination, axillary ultrasound, and biopsy (fine needle aspiration) in clinically suspicious nodes do not confirm metastases. This technique is also used in patients receiving neoadjuvant chemotherapy who did not have axillary lymph node metastases before treatment or in patients with metastases who achieved the clinical remission of nodal lesions (cN+/ycN0) [3,4,5,6,7].

Various techniques are used to identify the sentinel lymph node. Among them, the most common are radioisotopes, blue dyes, ferromagnetic approaches, and fluorescence [8,9]. Many clinical studies have demonstrated that no single method has a clear advantage over others in terms of effectiveness and the SLN detection rate [10,11,12,13,14,15,16,17,18].

The radioisotope method is well established as the gold standard due to its high precision and safety, even during pregnancy [19]. In this technique, a 99mTc-labelled colloid at a dose of 1 mCi (37 MBq) in 0.5 mL is prepared in a nuclear medicine laboratory. It is injected periareolarly or peritumorally 4 to 24 h before surgery. The efficacy of the injection is assessed through scintigraphy immediately after administration. The sentinel lymph nodes are detected intraoperatively using a handheld gamma radiation detector. It does not impact imaging modalities, ensures consistency in node identification, and does not cause skin pigmentation. However, it relies on the availability of nuclear medicine facilities, which may limit its accessibility in some centres.

In contrast, the SPIO method involves the injection of 1 mL of superparamagnetic iron oxide periareolarly or peritumorally, which can be administered as early as 20 min or up to 30 days before surgery [20]. The sentinel lymph nodes are identified using a magnetic sensing probe, with the added benefit of visual confirmation. However, this method has some drawbacks, including skin pigmentation, potential artefacts during MRI imaging, and contraindications in patients with iron overload disease or hypersensitivity to iron oxide [21,22]. Furthermore, its safety in pregnant or nursing women has not yet been confirmed.

This study compared the efficacy of SPIO and RI SLNB in breast cancer patients from two independent breast cancer centres. The identification rates and the number of collected and positive sentinel nodes (SLNs) were analysed in this study.

## 2. Materials and Methods

### 2.1. Study Group

We retrospectively reviewed data from 406 breast cancer patients who underwent SLNB between January 2021 and December 2022 in two cancer centres. In the Department of Surgical Oncology and Breast Diseases, 223 (54.9%) SLNBs were performed using the SPIO technique (Group A). In the Department of Surgical Oncology (Group B), 183 (45.1%) patients underwent SLNB using only the isotopic method.

In both groups, the inclusion criteria were breast cancer with clinically negative axillary lymph nodes (both qualified for primary surgery and after neoadjuvant treatment). We excluded patients who were pregnant or undergoing re-SLNB and cases of male breast cancer.

Along with the SLNB, each patient underwent breast surgery (BCT or mastectomy).

Further treatment processes were established during the MDT meeting, according to the current recommendations. All patients who qualified according to the applicable guidelines were subjected to adjuvant radiotherapy—all those after breast-conserving surgery, as well as patients after mastectomy with the T4 or N+ feature. The area of irradiation depended on the stage of advancement. In patients with the N+ feature, the irradiation area included the axillary, supraclavicular, and retrosternal nodes.

Table 1 presents the demographic and clinicopathological factors of the patients. A two-tailed Fisher’s exact test showed that the groups were statistically similar (*p* = 1.0).

The requirement for ethical committee approval was waived because of the study’s retrospective nature and the anonymisation of the clinical data.

### 2.2. Description of the Technique

In Group A, superparamagnetic iron oxide (SPIO; Magtrace^®^, Endomagnetics, Ltd., Cambridge, UK) was administered one day before the surgery. Each patient received a periareolar injection of 1 mL undiluted SPIO. The surgeon used a handheld probe (Sentimag^®^ G2, Endomagnetics, Ltd.) during the surgery to identify the SLN after the skin incision. At this time, all metal instruments were removed. A Sentimag^®^ magnetometer was used on setting 2 (Figure 1a). All nodes were collected until the counts fell below 10% of the highest count. 

Patients from Group B had a radiocolloid labelled with technetium-99m (99mTc) injected periareolarly at a dose of 1 mCi in 0.5 mL one day before surgery. The effectiveness of the injection was assessed immediately after administration using scintigraphy (Figure 2). A handheld gamma detector (Gamma Finder^®^ II; W.O.M. World of Medicine GmbH; standard operating mode) was used to locate the sentinel node during the procedure (Figure 1b).

### 2.3. Pathological Evaluation

All hot/brown lymph nodes and any palpable suspect nodes were collected and sent for examination. In Group A, most patients underwent an intraoperative SLN assessment using frozen sections. All patients with confirmed macro-metastasis at the frozen section underwent immediate ALND, while patients with positive SLNs at the definitive histopathological examination qualified for the second procedure. In Group B, all nodes underwent a routine pathological evaluation, and patients with positive SLNs underwent delayed ALND. Axillary lymph node dissection was performed at a similar frequency in both groups (chi-squared test, *p* = 0.61).

The groups were compared in terms of age, cancer stage, surgical treatment (breast-conserving or mastectomy with or without immediate breast reconstruction), and preoperative systemic treatment. We reviewed the average number of retrieved SLNs, the general number of retrieved SLNs, and metastatic lymph nodes dissected depending on the method used. 

## 3. Results

This study included 406 patients. There were no significant differences between the cohorts concerning their age, tumour size, tumour type, and biology. A two-tailed Fisher’s exact test showed that the groups were statistically similar (*p* = 1.0). The average patient age was 60.4 (57.7 in Group A and 63.6 in Group B). In both groups, most of the patients (95.1% in both groups) undergoing SLNB suffered from early-stage breast cancer (Tis/T1/T2). The most common histological type was invasive ductal cancer—74.4% of patients in Group A and 71.0% in Group B. In both groups, breast-conserving treatment prevailed over mastectomy in 64.1% and 74.3% of cases.

Neoadjuvant systemic treatment was administered in 31 (13.9%) and 9 (4.9%) patients, respectively. In the SPIO group, 11 patients had clinically positive nodes before systemic treatment. After neoadjuvant chemotherapy, all 11 patients presented clinical remission and qualified for SLNB. Only three of them had nodal metastases confirmed during the frozen section analysis and, therefore, required axillary lymph node dissection. In the RI group, six patients had confirmed metastases before chemotherapy. All of them presented clinical remission and underwent SLNB. Three patients required ALND due to exhibiting the ypN+ feature.

The SLN identification rates with the SPIO and RI groups were 99.6% (222/223) and 100% (183/183), respectively. The number of SLNs ranged from 1 to 10 in Group A and 1 to 12 in Group B (Figure 3).

The mean number of retrieved SLNs was significantly higher in Group A, with a mean value of 3.26, compared to 2.15 in Group B (Mann–Whitney U test, *p* < 0.001) (Figure 4).

There was no statistically significant difference between the groups in the number of patients with lymph node involvement based on the TNM classification—the N1 stage was found in 23.8% of patients in Group A vs. 21.3% in Group B (chi-squared test, *p* = 0.56) (Figure 5).

However, a statistically significant difference was observed in the proportion of metastatic SLNs to harvested SLNs between the techniques, with a larger proportion detected in the RI group (1/7.88) compared to the SPIO group (1/14.81) (chi-squared test, *p* < 0.03).

Additionally, a significantly larger proportion of patients in Group A had five or more SLNs retrieved than in Group B—20.2% vs. 8.7% (chi-squared test, *p* = 0.001) (Figure 6).

## 4. Discussion

In our study, the identification rate for sentinel lymph nodes was 99.6% (222/223) with the superparamagnetic iron oxide tracer and 100% (183/183) with the standard radiotracer 99mTc. These results are consistent with other studies evaluating SPIO as an alternative to radiotracers. Thill et al. reported identification rates of 98.0% for SPIO and 97.3% for 99mTc [23], while Rubio et al. observed a detection rate of 98.3% for SPIO and 95.7% for 99mTc, with a concordance rate of 98.2% [24]. The Sentimag Multicentre Trial also demonstrated comparable identification rates, with 94.4% for SPIO and 95.0% for the standard technique [25]. Although our results show a slightly higher identification rate with the radiotracer, the performance of the SPIO tracer remains robust and aligns well with the findings of these prior studies.

The optimal SLN number is still controversial. The mean number of SLNs collected ranges from 1.2 to 3.4, totalling 1 to 15 SLNs. The SPIO method in our study had a significantly higher average number of SLNs than the RI method. The patients in Group A (SPIO) had an average of 3.26 nodes removed, whereas the patients in Group B (RI) had only 2.15. The results of previous studies, such as the Sentimag trial and the study by Rubio et al., show that SPIO increases the number of nodes per patient. SPIO has been suggested to have a wider detection range, which could result in a higher nodal yield [24,25].

The sentinel lymph node is the primary node for lymphatic drainage from the tumour bed, but multiple SLNs may exist. The optimal number of removed SLNs has not yet been determined. When only a single SLN is removed, the main concern is the increase in the false negative rate (FNR). Ban et al. suggested that SLNB should not only collect one or two hot nodes when other hot nodes exist. They also recommend that four might be an optimal threshold number of SLNs to be removed and that removing more than four SLNs does not improve the axillary staging accuracy [26]. Similar conclusions were obtained by Chagpar et al. as they demonstrated that limiting SLNB to three nodes increases the FNR by 10.3% (*p* = 0.005 compared to removing > three SLNs) [27].

The question remains about the upper limit, as the risk of complications increases with the number of SLNs retrieved. The main benefit of de-escalating axillary surgery and introducing SLNB instead of ALND is reducing morbidity. SLNB significantly minimises the risk of lymphedema compared to ALND, and this is estimated at around 5%. The likelihood of developing lymphedema is higher when more than five SLNs are removed (3.7% of patients with more than five SLNs excised vs. 1.4% of patients with ≤ five SLNs excised) [28]. Other studies, like that of Yi et al., indicate that the removal of a maximum of five SLNs at surgery allows for the recovery of more than 99% of positive SLNs in patients with breast cancer, which confirms that it is not only potentially harmful but also unnecessary to retrieve more SLNs [29]. In our study, 20.2% of the patients in the SPIO group had more than five SLNs collected (compared to 8.7% in the RI group), representing a clear disadvantage of the magnetic tracer.

Furthermore, Cipolla et al. examined the relationship between the number of dissected SLNs and median disease-free survival (mDFS). Their results indicated no relevant difference in mDFS for patients with one SLN removed compared to those with more than one, except for a subset of patients treated with hormone therapy, where differences in the outcomes were observed. This highlights the need to explore patient-specific factors influencing SLNB outcomes [30].

When comparing the two methods, the RI method demonstrated a notable advantage in precision in our study. While retrieving a statistically smaller number of sentinel lymph nodes (SLNs) on average compared to the SPIO method (SPIO group: 3.26 vs. RI group: 2.15, *p* < 0.001)—an observation consistent with previously published studies [22,24,25]—it identified metastatic involvement in a larger proportion of the sampled nodes (RI group (1/7.88); SPIO group (1/14.81); chi-squared test, *p* < 0.03). Interestingly, this outcome was observed despite there being no significant difference between the groups regarding the disease stage or nodal involvement based on the TNM classification. This intriguing observation suggests a potential inherent advantage of the RI method in targeting clinically relevant nodes. However, this study’s retrospective design limits our ability to explain this phenomenon fully. Further prospective research is essential to confirm the reproducibility of this finding and investigate the mechanisms underlying this discrepancy.

Sentinel lymph node biopsy is also performed in patients with DCIS, e.g., in case of the presence of a palpable tumour (>2 cm in size), and the administration of RI or SPIO and the surgical procedure itself are carried out in the same manner as in patients diagnosed with invasive cancer [31,32]. The likelihood of finding metastases in the sentinel nodes of patients with a final histopathological diagnosis of DCIS is minimal; therefore, assessing the “accuracy” of metastasis detection is not relevant. Although the likelihood of metastasis is minimal, the choice of the sentinel node detection method may be crucial in avoiding excessive procedures, reducing the risk of complications, and ensuring that the assessed nodes accurately reflect the patient’s condition. In this context, SPIO presents a significant advantage. Because the tracer can be administered up to 30 days before surgery, patients diagnosed with DCIS can avoid unnecessary intervention in the axilla, and the sentinel nodes can be harvested later on, only if invasive cancer is confirmed on histopathology.

Patient-related factors should also be considered in the context of method optimisation. Lymphatic flow depends on age and body mass and is slower in obese and elderly patients. Makita et al. demonstrated that the injected dose of SPIO influences the number of SLNs. Increasing the tracer dose increased the number of SLNs, especially in younger patients with a lower BMI. Considering this study, adjusting the dose and the number of SLNs seems possible [33]. However, this issue does not arise with the radioisotope method, as the administered dose is precisely defined and does not depend on patient-related factors, representing a significant advantage of this method.

Additionally, surgical and treatment-related variables significantly affect the number of removed SLNs. According to Dixon et al., a multivariate analysis found that the operating surgeon, neoadjuvant chemotherapy (NAC), and the number of involved lymph nodes all significantly affected the number of SLNs collected. The removal of more nodes was more likely in patients who underwent NAC and had metastases in the sentinel lymph nodes. It can be inferred that both the disease burden and preoperative treatments influence the amount of nodal retrieval. These findings highlight the significance of tailoring sentinel node procedures according to individual patient characteristics and clinical contexts [34].

Patients with initially positive axilla undergoing neoadjuvant chemotherapy require specific consideration. In the past, these patients qualified for axillary lymph node dissection, regardless of the clinical and pathological response. Clinical trials, like Z1071, Sentina, and SN FNAC, changed this approach and allowed for the de-escalation of axillary surgery. These studies have proven that correctly performing SLNB reduces the FNR [6,35,36]. Factors that further reduce the FNR are collecting at least three SLNs and implementing targeted axillary dissection—marking the positive node before NAC and collecting it along with the SLN [37,38,39,40]. 

Using a single method is considered insufficient for sentinel lymph node identification in patients who have undergone chemotherapy. Therefore, two techniques—blue dye and a radioisotope—are recommended for concurrent application [41,42]. However, in both study groups, SLNB was performed using the single tracer method, without adding blue dye staining. In our study, in Group A, consisting of 31 patients who underwent chemotherapy, only superparamagnetic iron oxide (SPIO) was used as an independent sentinel lymph node identification method. In Group B, where the radioisotope-only method was employed, nine patients had received neoadjuvant chemotherapy. In these cases, blue dye was not administered due to adequate gamma radiation or magnetic uptake, which allowed the surgeon to determine intraoperatively that a satisfactory number of lymph nodes had been biopsied. The systematic review by He at al. concluded that, despite the fact that the combination of a radioisotope and blue dye has shown a higher identification rate overall, no significant advantage was observed in patients after neoadjuvant chemotherapy or those with high lymphoscintigraphy positivity. Additionally, dual tracers did not significantly reduce the false negative rate compared to the radioisotope method alone [43,44]. Moreover, according to Harper et al., blue dye accounts for 5% of intraoperative anaphylactic reactions [44,45]. In addition, scientific reports support the use of SPIO as a standalone method for sentinel lymph node detection in patients following chemotherapy. Studies indicate that SPIO offers a detection rate comparable to that of the dual-tracer technique, reducing logistical challenges and adverse effects such as allergic reactions [46]. It raises the question of whether it is necessary to expose patients who have undergone neoadjuvant chemotherapy to the risk of an anaphylactic reaction, especially when the radioisotope-only method and the SPIO method alone appear to be sufficient.

Breast cancer treatment is de-escalating, and SLNB is an excellent example of this, as it allows, in many cases, the avoidance of extensive and often burdensome axillary lymph node dissection. As a basic procedure, it should be perfected to be clinically significant, with a minimal risk of complications for the patient at the same time.

There are some limitations that affect our study. Firstly, the study was carried out at two separate cancer centres. Although the two groups of patients had no differences in terms of clinicopathological features, it is possible that the surgeons or surgical teams affected the results. Secondly, our investigation did not explore whether the number of sentinel nodes collected corresponded with post-surgical axillary lymphedema, which may be essential in determining the optimal technique of SLNB. Recurrence rates and the incidence of post-surgical complications are essential factors in evaluating the clinical applicability of SLNB methods. Therefore, randomised controlled trials are crucial in validating our findings and advancing the field of breast cancer treatment.

## 5. Conclusions

This study underscores the effectiveness of both SPIO and RI methods for sentinel lymph node biopsy (SLNB) in breast cancer patients, with high identification rates achieved using each approach.

The RI method allows highly targeted node identification, minimising the risk of over-dissection and ensuring that only the most clinically relevant nodes are removed. Furthermore, the RI method’s reliance on nuclear medicine facilities ensures standardisation and precision, making it an exceptionally reliable choice for centres with the necessary infrastructure.

The SPIO method demonstrates comparable efficacy to RI and provides logistical benefits, such as flexible administration timing and not requiring nuclear medicine facilities. However, it is associated with retrieving a larger number of sentinel lymph nodes (SLNs), which may increase the risk of patient morbidity, including complications such as lymphedema. While the RI method remains the benchmark for SLNB, we note that the SPIO technique requires further standardisation.

## Figures and Tables

**Figure 1 jcm-14-03148-f001:**
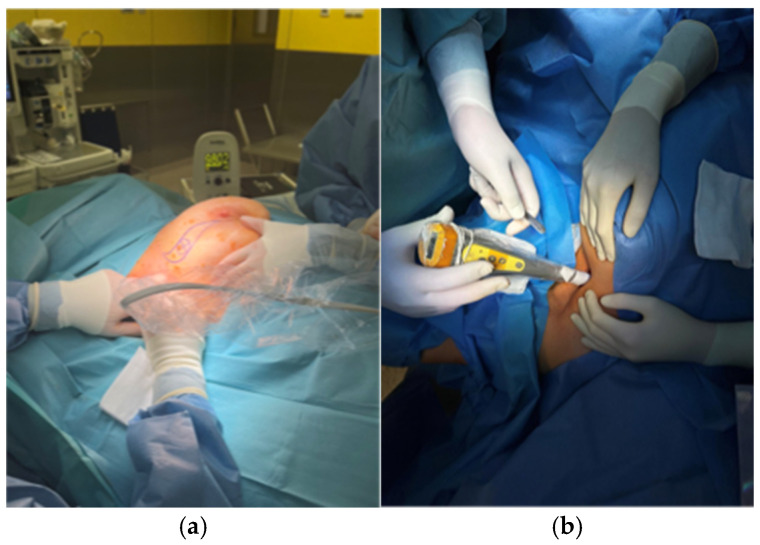
Intraoperative SLN identification using Sentimag^®^ probe (**a**) and Gamma Finder^®^ II probe (**b**).

**Figure 2 jcm-14-03148-f002:**
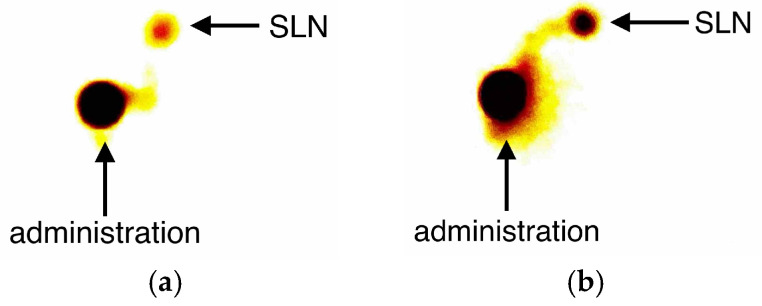
Result of scintigraphy performed immediately after RI administration in two projections—anterior–posterior (**a**) and side (**b**).

**Figure 3 jcm-14-03148-f003:**
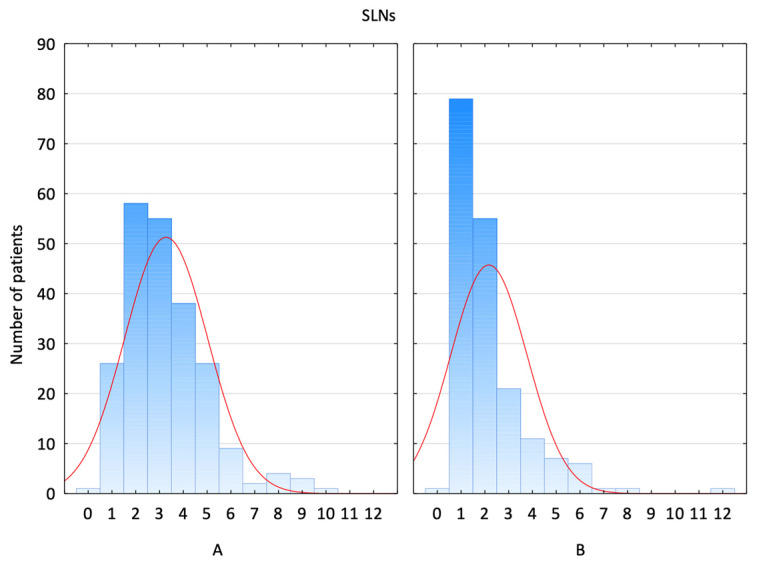
Distribution of SLNs retrieved. (A—SPIO group; B—RI group).

**Figure 4 jcm-14-03148-f004:**
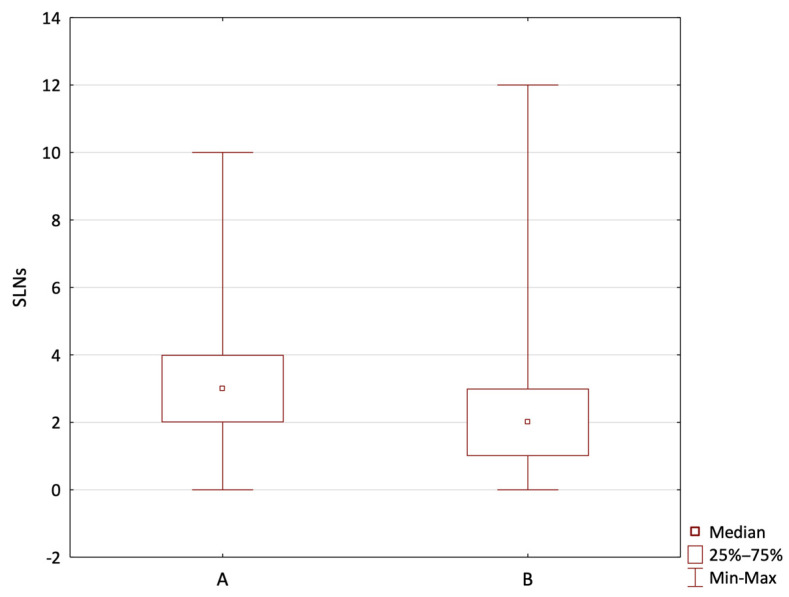
The mean number of retrieved SLNs. (A—SPIO group; B—RI group).

**Figure 5 jcm-14-03148-f005:**
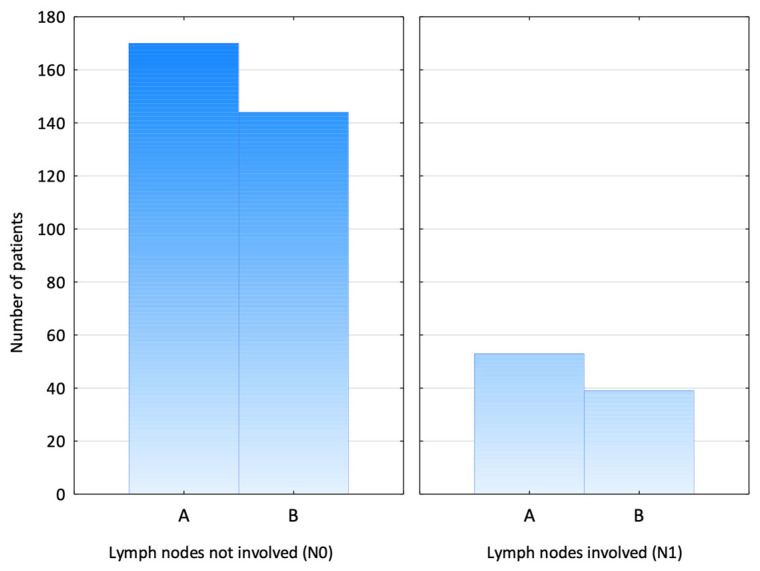
Nodal staging. (A—SPIO group; B—RI group).

**Figure 6 jcm-14-03148-f006:**
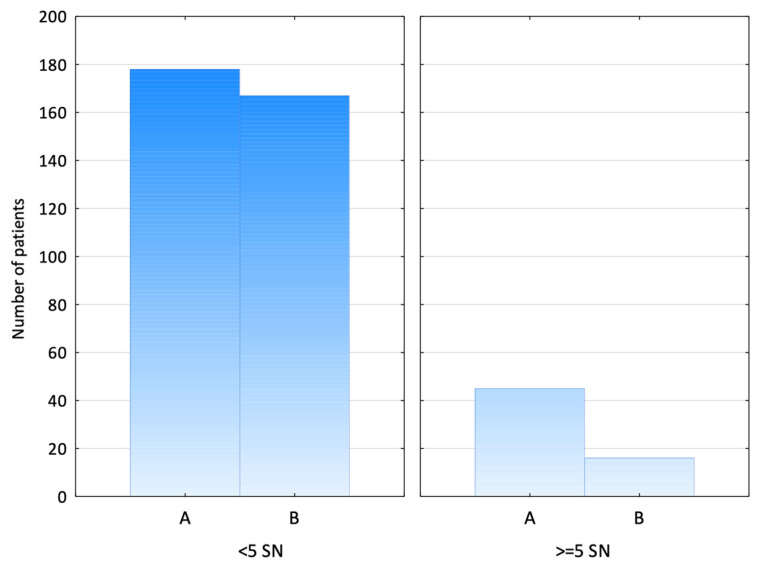
Patients with more than 5 SLNs retrieved. (A—SPIO group; B—RI group).

**Table 1 jcm-14-03148-t001:** Patients’ characteristics ^1^.

	Group A—SPIO	Group B—RI	Total
Patients	223	183	406
Age			
mean	57.7	63.6	60.4
<50 years old	74 (33.2)	42 (23.0)	116
>50 years old	149 (66.8)	141 (77.0)	290
Tumour size			
Tis	17 (7.6)	8 (4.4)	25
T1	103 (46.2)	96 (52.4)	199
T2	92 (41.3)	70 (38.3)	162
T3	7 (3.1)	8 (4.4)	15
T4	4 (1.8)	1 (0.5)	5
Type of surgery			
BCT	143 (64.1)	136 (74.3)	279
mastectomy	80 (35.9)	47 (25.7)	127
with IBR	40	0	40
Neoadjuvant chemotherapy	31 (13.9)	9 (4.9)	40
Histological type			
DCIS	17 (7.6)	8 (4.4)	25
IDC	166 (74.4)	135 (73.8)	301
Other	40 (17.9)	40 (21.8)	80

^1^ Data are presented as n (%) unless otherwise noted. BCT—breast-conserving therapy, IBR—immediate breast reconstruction, DCIS—ductal cancer in situ, IDC—invasive ductal cancer.

## Data Availability

The raw data supporting the conclusions of this article will be made available by the authors upon request.

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
