# Peer review of "Is the Superparamagnetic Approach Equal to Radioisotopes in Sentinel Lymph Node Biopsy? The Over-Collecting Node Issue in Breast Cancer Patients"

_jcm, 2025, doi:10.3390/jcm14093148_

Round 1
Reviewer 1 Report
Comments and Suggestions for Authors
I have read with great interest article entitled: Is superparamagnetic as good as radioisotope in sentinel lymph node biopsy? - the over-collecting nodes issue in breast cancer patients. This study compared the efficacy of SPIO and RI SLNB from two independent breast cancer centres. Based on the obtained results, SPIO and RI methods effectively collected SLNs. The gold standard RI method offers distinct advantages, including its precise and consistent dosing unaffected by patient-specific factors and a highly targeted approach to node identification. These features minimize the risk of over-dissection and ensure that only the most clinically relevant nodes are removed. Authors stated that the SPIO technique in SLNB in breast cancer patients requires further standardization. This is interesting article analysing two methods in detecting sentinel lymph nodes in clinically negative axilla of breast cancer patients. Basically, this is a negative study which proved that standard of care is superior compared to SPIO. While SPIO can identify some new information on collected LN there is no clinical relevance of these findings. Also, there is no information on recurrence rates after both methods performed. Additionally, how many patients received adjuvant radiotherapy and which areas of axilla were irradiated? There is also no information on postsurgical lymphedema in both groups. Furthermore, DCIS should be excluded from statistical analysis. How many patients received neoadjuvant therapy and what were the rates of conversion of cN+ in cN0? There is no clinical utility of these findings and obtained results cannot improve the current clinical practice for breast cancer.
Author Response
1. There is no information on recurrence rates after both methods performed.
Thank you for a thorough review of our manuscript and your thoughtful remarks. Our study was retrospective in nature. We analysed patients’ medical records available at the centres conducting the survey. These centres are major surgical institutions in their respective regions but are not the only facilities that provide oncological (surgical and systemic) treatments. For this reason, some of the patients included in the study continued their breast cancer treatment at other institutions (e.g., closer to their place of residence) or chose different hospitals in case of the need for reoperation, for example, due to recurrence. All this means that we do not have access to the complete medical history of the patients included in our analyses after their hospitalisation at our centre. Consequently, we can’t analyse recurrence rates or the occurrence of lymphorrhea. We include this information in the Discussion, paragraph “limitations”.
2. Additionally, how many patients received adjuvant radiotherapy and which areas of axilla were irradiated?
All patients who qualified according to the applicable guidelines were subjected to adjuvant radiotherapy - all those after breast-conserving surgery, as well as patients after mastectomy with the T4 or N+ feature. The area of ​​irradiation depended on the stage of advancement. In patients with the N+ feature, the irradiation area included the axillary, supraclavicular and retrosternal nodes. We added that information in the Material and Methods sections.
3. There is also no information on postsurgical lymphedema in both groups.
The study was retrospective, and due to its nature, it’s impossible to establish morbidity rates after sentinel node biopsy in our cohorts. However, this procedure was designed to minimise surgical intervention in the axilla, and one of its primary goals was to avoid lymphedema. The lack of observation regarding the postoperative lymphedema was emphasized in the Discussion section.
4. Furthermore, DCIS should be excluded from statistical analysis.
Our study aimed to compare the number of lymph nodes removed during the SLNB procedure depending on the method used for sentinel node detection. This procedure is also performed in patients with DCIS, e.g, in case of the presence of palpable tumour (>2cm in size), when the administration of RI or SPIO and the surgical procedure itself are carried out in the same manner as in patients diagnosed with invasive cancer.
We agree that the likelihood of finding metastases in the sentinel node in patients with a final histopathological diagnosis of DCIS is minimal, and therefore, assessing the "accuracy" of metastasis detection is not relevant. However, the primary goal of our study was to determine whether the use of either method (RI or SPIO) leads to unnecessarily extensive axillary procedures. Our findings suggest that fewer nodes are removed when using RI, and they are those that should be evaluated.
Is it reasonable, then, to include patients with DCIS? Even though their likelihood of metastasis is minimal, the choice of sentinel node detection method may be crucial in avoiding excessive procedures, reducing the risk of complications, and ensuring that the assessed nodes accurately reflect the patient’s condition.
Moreover, in this context, SPIO presents an advantage. Because the tracer can be administered up to 30 days before surgery, patients diagnosed with DCIS can avoid unnecessary intervention in the axilla, and sentinel nodes can be harvested later on, only if invasive cancer is confirmed on histopathology. We have extended the Discussion with a paragraph focused on the DCIS and SNB issue.
5. How many patients received neoadjuvant therapy and what were the rates of conversion of cN+ in cN0?
In our centres, patients with clinical multiple metastases to lymph nodes before systemic treatment were not subjected to sentinel lymph node biopsy; instead, they were qualified for axillary lymph node dissection and, therefore, were excluded from the study.
In our study, 40 patients received neoadjuvant chemotherapy — 31 (13.9%) in the SPIO group and 9 (4.9%) in the RI group. In the SPIO group, 11 patients had clinically positive nodes before systemic treatment. After neoadjuvant chemotherapy, all 11 patients presented clinical remission and were qualified for SLNB. Only 3 of them had nodal metastases confirmed during the frozen section and, therefore, required axillary lymph node dissection. In the RI group, 6 patients were cN+ before chemotherapy. All of them presented clinical remission and had SLNB done. 3 patients required ALND because of the ypN+ feature. The group of patients receiving neoadjuvant treatment is small, presenting a certain study limitation. We added the data on the conversion of cN+ to cN0.
Reviewer 2 Report
Comments and Suggestions for Authors
It’s a well written manuscript entitled “the investigating the role of Primary procedure for nodal assessment in Sentinel lymph node biopsy in breast cancer patients”.
The research study involves the data collection from 406 breast cancer patients who suffered and were diagnosed with SLNB.
The story is relevant to clinical researchers and cancer biologists which explores the importance of two methods of gold standard RI methods and SPIO techniques in SLNB in breast cancer patients with variations in identification rates.
The present study approach is having much significant information about the role of both techniques in variable administration timing and autonomy from the nuclear medicine facility.
This manuscript should have minor revision to increase the clarify the specific feedback:
- There are many challenges with the complexity of the malignancy related to the sentinel node biopsy that is not accurate in chemotherapy. So, how did authors exclude this false negative approach and artifacts in this study to choose the patients?
- SLN is mostly dependent upon the stage of the breast cancer and biopsy is justified because the size of the tumor mass could potentially affect the treatment result, so how do authors address this point? Also, do authors include the consequence of SLN Biopsy related symptoms in high mortality and how it influences the precision of the diagnostic approach in this study?
- Unlike RI methods, SPIO method in SLNB does not have a radiation exposure, although there is a limitation in efficacy in specific patient’s groups and dose optimization related issues. How do authors answer this question?
- Authors should elaborate more about the significance and therapeutic part of this study.
Thanks
Author Response
1. There are many challenges with the complexity of the malignancy related to the sentinel node biopsy that is not accurate in chemotherapy. So, how did authors exclude this false negative approach and artifacts in this study to choose the patients?
Thank you for your detailed review of our manuscript and your thoughtful comments. The MDT meetings establish qualification for the SNB or ALND and the entire treatment process. All decisions are supported by the consensus of Polish oncological societies' recommendations and NCCN and ESMO guidelines. In complex situations, e.g., in patients cN1 who received preoperative chemotherapy, lymph nodes ultrasound is routinely performed to confirm complete clinical response. Such an approach reduces the risk of false negative results.
2. SLN is mostly dependent upon the stage of the breast cancer and biopsy is justified because the size of the tumor mass could potentially affect the treatment result, so how do authors address this point?
It is believed that the likelihood of developing metastases increases with tumour size. The vast majority of patients qualified for SLNB in both groups had early-stage breast cancer (Tis/T1/T2), which would seem to confirm this. However, the relationship between tumour size and nodal and distant metastases in patients with invasive breast cancer is not linear, and the risk of metastasis is dependent mainly on the tumour biology rather than the size. An undoubted benefit of preoperative chemotherapy is the clinical remission of breast and axillary lesions, which allows, in many cases, to omit the axillary lymph node dissection in favour of sentinel node biopsy or targeted axillary dissection, when applicable. We addressed this issue in the Discussion and added the references.
3. Also, do authors include the consequence of SLN Biopsy related symptoms in high mortality and how it influences the precision of the diagnostic approach in this study?
Lymphedema usually develops several months or years after surgery and radiotherapy. Long-term observation would be needed to evaluate this complication. As the risk of complications increases with the increasing number of collected nodes, patients with more than 5 nodes collected would require detailed assessment. We emphasised this issue in limitations.
4. Unlike RI methods, SPIO method in SLNB does not have a radiation exposure, although there is a limitation in efficacy in specific patient’s groups and dose optimization related issues. How do authors answer this question?
More studies address the administered dose issue, but there are still no clear guidelines. As lymphatic flow is associated mainly with the BMI and the patient's age, attempts are being made to lower the dose in younger people and those with a lower BMI. We re-emphasized this issue in the Discussion and added the reference.
5. Authors should elaborate more about the significance and therapeutic part of this study
While including all remarks pointed out by the Reviewers in the manuscript, we believe that the therapeutic significance of our study becomes more emphasised. However, we added a comment in the Discussion that sentinel lymph node biopsy is an excellent example of de-escalation in breast cancer surgery, as it allows, in many cases, to avoid extensive and often burdensome axillary lymph node dissection. As a basic procedure, it should be perfected to be clinically significant with minimal risk of complications for the patient at the same time.
Round 2
Reviewer 1 Report
Comments and Suggestions for Authors
Although authors addressed most of the comments, there are no data on recurrence rates during follow-up period and postsurgical lymphoedema which are crucial information in assessing the applicability of this method in clinical practice.
Author Response
Our study was conducted retrospectively, which limited our access to the comprehensive medical histories of all patients after their hospitalization at our centres. Certain patients who pursued their breast cancer treatment at different institutions impeded our ability to gather thorough follow-up data regarding recurrence rates and post-surgical lymphedema. Consequently, our study does not allow for a direct assessment of these critical outcomes.
We acknowledge that recurrence rates and the incidence of post-surgical lymphedema are essential factors for evaluating the clinical applicability of SLNB methods. However, addressing these concerns would require a prospective study design characterised by long-term follow-up and active patient monitoring. Our primary objective was to compare the number of lymph nodes removed utilizing various detection methods (RI vs. SPIO), thereby yielding valuable insights into the extent of axillary surgery.
Similar retrospective studies have been conducted to compare different SLN detection methods, including the study by Yang and Zhang, which examined Indocyanine green combined with methylene blue versus methylene blue alone for sentinel lymph node biopsy in breast cancer: a retrospective study. Their research retrospectively analyzed the effectiveness of combining indocyanine green (ICG) with methylene blue (MB) compared to MB alone in SLNB. The study revealed that combining ICG and MB improved sentinel lymph node detection rates and facilitated the identification of a greater number of lymph nodes. Our study focused on comparing the efficacy of different tracers without assessing long-term oncological outcomes such as recurrence rates or post-surgical complications. This further emphasizes that retrospective analyses primarily provide insight into procedural efficiency and feasibility. At the same time, prospective studies with long-term follow-up remain indispensable for comprehensively evaluating the clinical impact of these methods.
Future prospective studies with extended follow-up periods will be essential to thoroughly evaluate these techniques’ long-term oncological outcomes and morbidity.